# A combinatorial action of GmMYB176 and GmbZIP5 controls isoflavonoid biosynthesis in soybean (*Glycine max*)

Arun Kumaran Anguraj Vadivel [1,2], Tim McDowell [1], Justin B. Renaud[1] & Sangeeta Dhaubhadel [1,2✉]

GmMYB176 is an R1 MYB transcription factor that regulates multiple genes in the iso-flavonoid biosynthetic pathway, thereby affecting their levels in soybean roots. While GmMYB176 is important for isoflavonoid synthesis, it is not sufficient for the function and requires additional cofactor(s). The aim of this study was to identify the GmMYB176 inter-actome for the regulation of isoflavonoid biosynthesis in soybean. Here, we demonstrate that a bZIP transcription factor GmbZIP5 co-immunoprecipitates with GmMYB176 and shows protein–protein interaction *in planta*. RNAi silencing of *GmbZIP5* reduced the isoflavonoid level in soybean hairy roots. Furthermore, co-overexpression of *GmMYB176* and *GmbZIP5* enhanced the level of multiple isoflavonoid phytoalexins including glyceollin, isowighteone and a unique O-methylhydroxy isoflavone in soybean hairy roots. These findings could be utilized to develop biotechnological strategies to manipulate the metabolite levels either to enhance plant defense mechanisms or for human health benefits in soybean or other eco-nomically important crops.

[1] London Research and Development Centre, Agriculture and Agri-Food Canada, London, ON, Canada. [2] Department of Biology, University of Western Ontario, London, ON, Canada. ✉email: sangeeta.dhaubhadel@canada.ca

soflavonoids are biologically active legume-specific specialized metabolites with pharmacological properties[1]. They play an important role in the interaction between plants and their environment. Isoflavonoids act as chemoattractants to rhizobia and facilitate their symbiotic relationship with legume plants[2]. In response to pest and pathogen attack, soybean plants produce isoflavonoid phytoalexins that inhibit pathogen growth and provide broad resistance against them[3–5].

Soybean seeds contain three main isoflavone aglycones (genistein, daidzein, and glycitein) and their corresponding glycosides and malonylglycosides. Isoflavonoids are derived from the central flavanone intermediates naringenin and liquiritigenin, which in turn are derived from tetrahydroxychalcone (naringenin chalcone) and trihydroxychalcone (isoliquiritigenin chalcone), respectively. The enzyme Chalcone synthase (CHS) is involved in the condensation of p-coumaroyl-CoA with three acetate moieties, derived from malonyl-CoA, to form naringenin chalcone, and is the first step in the branched pathway for the synthesis of flavonoids and isoflavonoids[6]. Soybean contains 14 GmCHS genes (GmCHS1–GmCHS14) that play various roles during plant development or in response to environmental stimuli[7,8]. The members of GmCHS family show differential temporal and spatial expression. Among them, GmCHS7 and GmCHS8 are widely studied as GmCHS8 transcript abundance is directly associated with isoflavonoid levels in soybean seeds[9]. Furthermore, the transcript level of GmCHS7/GmCHS8 in seed coats determines yellow or black color soybean[10].

In eukaryotes, transcriptional regulation is often mediated by multi-protein complex or the concerted action of several proteins. Such proteins are part of an interactome where members of the complex may bind DNA directly or facilitate the interaction of other proteins within the complex. For example, the interaction of bZIP and Dof transcription factors regulate GST6 and 22-kDa class of zein genes in Arabidopsis[11] and maize[12], respectively. The protein complex containing maize C1 and R transcription factors has been shown to regulate anthocyanin biosynthesis in Arabidopsis and tobacco[13]. Genes involved in flavonoid biosynthesis are well conserved in higher plants and are regulated by a combinatorial action of transcriptional regulatory factors expressed in temporal and spatially controlled fashion[14,15]. The expression of early biosynthetic genes involved in flavonoid biosynthesis, such as Phenylalanine ammonia-lyase, CHS, Chalcone isomerase (CHI), Flavonol 3′-hydroxylase, Flavonol synthase (FLS) is regulated by MYB transcription factors in a coordinated manner[16] while the late biosynthetic genes are regulated by an MBW ternary complex consisting of a R2R3 MYB transcription factor, a basic helix-loop-helix (bHLH) transcription factor and a WD repeat protein[17,18]. Previously we discovered that the expression of GmCHS8 and isoflavonoid biosynthesis is regulated by an R1 MYB transcription factor GmMYB176[19]. Using the transcriptomic and metabolomic analysis, we uncovered that GmMYB176 regulates multiple steps in isoflavonoid biosynthesis[20]. Furthermore, detailed functional analysis of GmMYB176 revealed that it requires additional factor(s) such as another transcription factor or enhancers/repressors or a scaffold protein to activate GmCHS8 gene expression and isoflavonoid biosynthesis[19].

In this study, we identified GmMYB176 interacting factors, validated their protein–protein interaction in planta, and determined their DNA binding activity. RNAi silencing of the GmMYB176 interacting candidates, GmbZIP4 and GmbZIP5, and overexpression of the translational fusion of GmMYB176-GmbZIP4 and GmMYB176-GmbZIP5 in soybean hairy roots identified GmbZIP5, a basic leucine zipper family protein, as an interacting partner of GmMYB176 with a role in isoflavonoid biosynthesis. Our results demonstrate that both GmMYB176 and GmbZIP5 are co-expressed in soybean roots and their combined

action is critical to activate isoflavonoid biosynthesis in soybean roots.

## Results

**Identification of GmMYB176-interacting proteins.** To identify the GmMYB176 interactome in soybean, translational fusions of GmMYB176 with a yellow fluorescent protein (YFP) at either the N- or C-terminal (YFP-GmMYB176 or GmMYB176-YFP) were created and overexpressed in soybean hairy roots. The fusion proteins were created to use YFP as a tag in the co-immunoprecipitation (Co-IP) experiments. Despite the YFP tag position, both GmMYB176-YFP and YFP-GmMYB176 were localized to the nucleus and the cytoplasm (Fig. 1a).

GmMYB176 interacting proteins from soybean hairy roots overexpressing either GmMYB176-YFP or YFP-GmMYB176 were precipitated in two separate Co-IP experiments. The presence of the bait in the crude protein sample and in the eluate was confirmed by Western blot analysis (Fig. 1b). A total of 802 proteins were identified in the eluate of all three replicates for both GmMYB176-YFP and YFP-GmMYB176 fusion protein baits. Previously, we showed that some soybean hairy root proteins interact with YFP, and co-elute with it in Co-IP[21]. Therefore, to remove non-specific YFP interactors and to obtain GmMYB176-specific interacting candidates, YFP-interacting proteins from soybean hairy roots were subtracted from the list containing GmMYB176-YFP and YFP-GmMYB176 interacting proteins (Fig. 1c). This process identified a total of 716 candidate proteins where 105 candidates were common in both GmMYB176-YFP and YFP-GmMYB176 fusion baits (Fig. 1c, Supplementary Data 1). The candidates identified exclusively with GmMYB176-YFP (242 proteins) or YFP-GmMYB176 (369 proteins) were also included in the study as it is possible that some interactors may have been missed in one of the baits due to the position of YFP in the fusion protein. The biological activity and domain enrichment of the 716 candidate proteins were retrieved from GO annotation[22], and grouped into the categories based on their biological process, cellular component, and molecular function (Fig. 1d). The biological process—flavonoid biosynthesis included four proteins, GmCHI1A, GmCHI1B1, GmCHI4A, and GmCHS14 (Fig. 1d). Since GmCHI and GmCHS are not transcription factors, we focussed our efforts on the proteins that have putative DNA binding ability. Twenty-nine putative transcription factors belonging to 21 families were retrieved from GO annotation-Biological process-Transcription, DNA-dependent. Additionally, we performed in silico analysis of 30 bp GmCHS8 promoter region (23 bp with 7 bp flanking sequence) for regulatory elements binding sites as this region is critical for GmMYB176-mediated gene expression[19]. This process uncovered 23 transcription factors belonging to six families (Supplementary Data 2). Comparison of candidate transcription factors obtained through these two analyses identified two transcription factor families: bZIP [Glyma.04G222200 (GmbZIP4) and Glyma.05G122400 (GmbZIP5)] and R1 MYB (Fig. 2a). As a component of the MBW ternary complex, MYB and bHLH transcription factors have been shown to regulate flavonoid biosynthesis in many plants[18]. Therefore, three bHLH proteins [Glyma.05G134400 (GmbHLH5), Glyma.15G005100 (GmbHLH15), and Glyma.07G205800 (GmbHLH7)] were also chosen from the Co-IP list for the validation of their protein-protein interaction with GmMYB176.

To validate the GmMYB176 interacting candidates, bimolecular fluorescence complementation (BiFC) assay was used[23]. The assay was conducted using split YFPs, where translational fusions of N- or C-terminal halves of YFP were fused with the two proteins under investigation and transiently co-expressed in Nicotiana

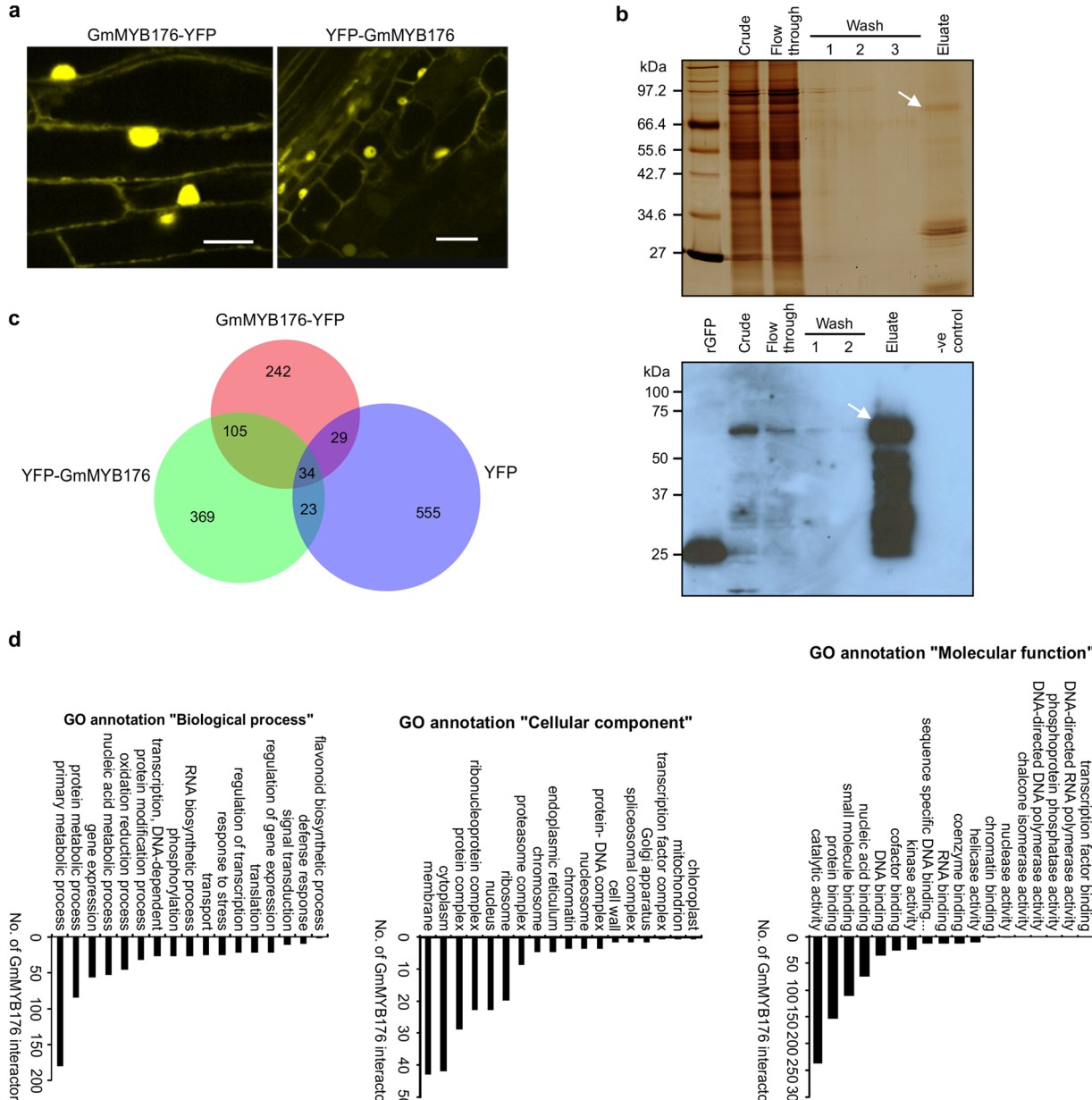

**Fig. 1 Co-immunoprecipitation of GmMYB176 interacting proteins from soybean hairy roots. a** Subcellular localization of GmMYB176-YFP and YFP-GmMYB176 in hairy roots. Both GmMYB176-YFP and YFP-GmMYB176 fusion proteins were localized in the nucleus and the cytoplasm of soybean hairy root cells as observed by confocal microscopy. Scale bar = 50 μm. **b** Crude protein extracts were subjected for Co-IP assay using anti-GFP microbeads and μMAC epitope tag protein isolation system. Samples from each step were separated on an SDS-PAGE and visualized by silver staining (top gel). The bottom image shows Western blot analysis using anti-GFP monoclonal antibody. The arrow indicates the estimated size of GmMYB176-YFP protein in the eluate. Crude: crude protein extract from soybean hairy roots; Flow through: crude extract incubated with anti-GFP microbeads and applied to μcolumn, with the flow through collected; Wash: sequential wash steps with lysis buffer; Eluate: elution of bound proteins from the column; –ve control: crude extract from control hairy roots. **c** Venn diagram showing the overlap of GmMYB176-YFP, YFP-GmMYB176, and YFP-only interacting candidate proteins in soybean hairy roots identified by LC–MS/MS analysis. The YFP interacting protein candidates were obtained from our previous study[21]. **d** 'GO' annotations of the 716 candidate GmMYB176-interacting proteins. List of soybean genes encoding the candidate proteins was used in PhytoMine[22] to generate annotations regarding the biological process, cellular component, and the molecular function of the candidates.

*benthamiana* leaves. As shown in Fig. 2b, the BiFC assay confirmed that GmMYB176 interacts with GmbZIP4, GmbZIP5, GmbHLH5, and GmbHLH15 in the nucleus. However, no interaction was observed between GmMYB176 and GmbHLH7 *in planta*. We previously demonstrated that GmMYB176 is a phosphoprotein and its phosphorylation state determines protein–protein interaction[24]. Therefore, protein–protein interaction using the phospho-site mutant of GmMYB176, GmMYB176S29A, was also performed.

The results demonstrated that GmMYB176S29A interacts with GmbZIP4 and GmbZIP5 in the nucleus. Based on the intensity of fluorescence, the interaction of GmMYB176S29A with GmbZIP4 and GmbZIP5 appeared stronger compared to their interaction with GmMYB176 (Fig. 2b). Furthermore, GmMYB176S29A did not interact with GmbHLH5 and GmbHLH15 *in planta*.

GmMYB176 interactor(s) involved in *GmCHS8* gene activation must possess the ability to bind *GmCHS8* promoter. Therefore,

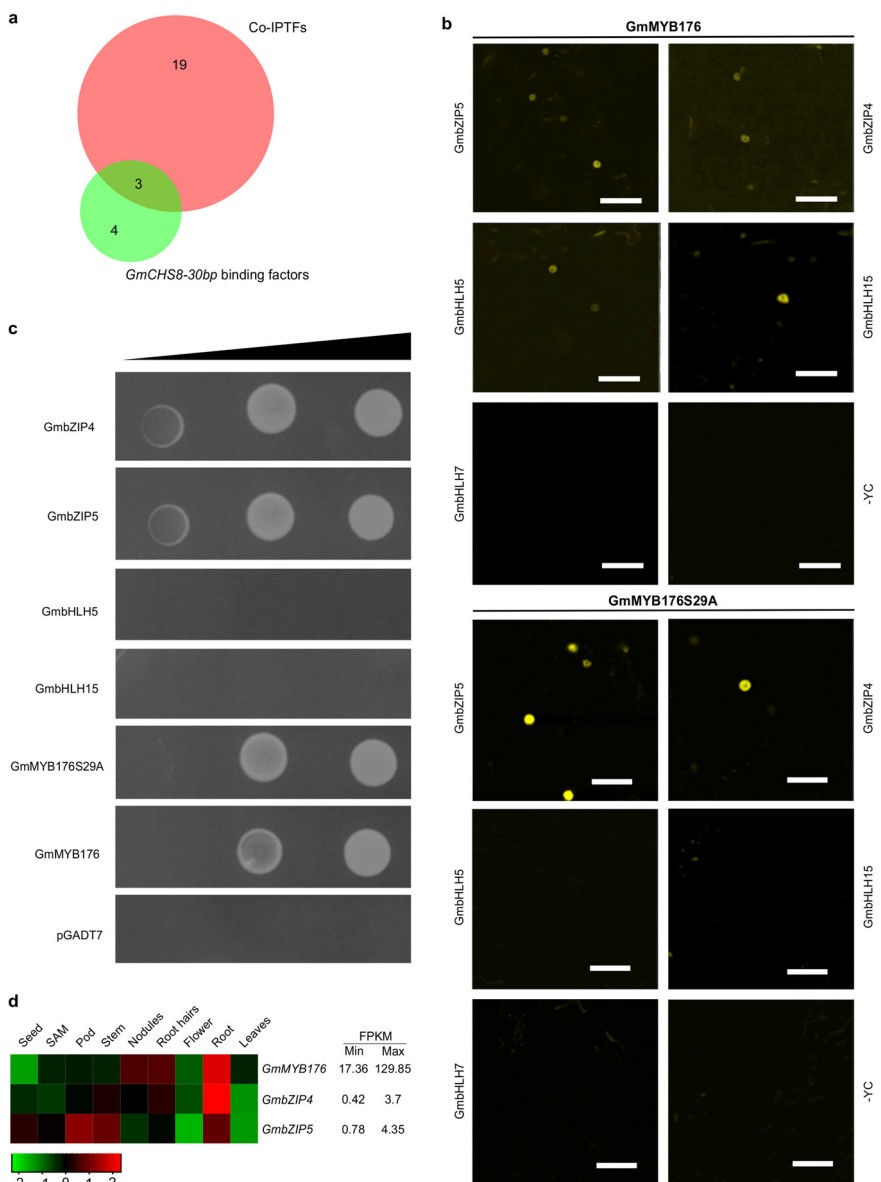

**Fig. 2 GmMYB176-interacting transcription factors and their DNA binding activity. a** GmMYB176- transcription factors obtained by co-IP assay GO annotation "Biological process" was retrieved and compared with the list of transcription factors obtained from *in silico* analysis of 30 bp *GmCHS8* promoter fragment (Supplementary Table 2) using a Venn diagram. **b** Protein–protein interactions of GmMYB176 and GmMYB176S29A *in planta* with candidate transcription factors obtained by Co-IP. The interaction between the proteins was assayed by co-expression of translational fusions of candidate proteins with N-terminal (YN) and C-terminal (YC) halves of YFP. The proximity of the two fragments results in a functional fluorophore. The fluorescence indicates the presence and location of the interaction between GmMYB176 or GmMYB176S29A with the candidate transcription factors. Fluorescent intensity parameters were kept constant in all images. Scale bars = 50 μm. **c** *GmCHS8* promoter (30 bp fragment) binding activity of GmMYB176 interacting candidates. Yeast cells carrying 30 bp *GmCHS8* tandem repeats (30 bpTR) as a bait, were transformed with prey constructs fused to a GAL4 activation domain. Growth on SD lacking leucine and in the presence of Aureobasidin A (SD/-Leu/AbA) shows the activation of the reporter and indicates DNA binding activity. As a negative control, pGADT7 vector only was used. **d** Tissue-specific expression pattern of *GmMYB176*, *GmbZIP4*, and *GmbZIP5* in soybean. RNA-seq data across different tissues were extracted from soybean whole genome database in Phytozome (https://phytozome.jgi.doe.gov/pz/portal.html#!info?alias=Org_Gmax) and a heatmap was constructed. Numbers to the right indicate the maximum value of fragments per kilobase of million mapped reads (FPKM). The color scale indicates expression values, green indicating low transcript abundance, and red indicating high levels of transcript abundance.

the GmMYB176 interactors validated *in planta* were assessed for their DNA binding ability using 30 bp tandem repeat of *GmCHS8* promoter (GmCHS8-30bpTR) in a yeast one-hybrid (Y1H) assay. The result revealed that both GmbZIP4 and GmbZIP5 bind with GmCHS8-30bpTR (Fig. 2c, Table 1). Even though GmbHLH5 and GmbHLH15 showed protein–protein interaction with GmMYB176, they lacked GmCHS8-30bpTR binding activity. GmMYB176 and empty prey vector (pGAD7) were used as

positive and negative controls, respectively. Since relatively stronger protein–protein interaction was observed between GmMYB176S29A and GmbZIP4/GmbZIP5 compared to GmMYB176 and GmbZIP4/GmbZIP5, we also examined the DNA binding ability of GmMYB176S29A and discovered that the DNA binding activity of GmMYB176 does not depend on its phosphorylation state. Based on the protein–protein and protein–DNA interactions (Fig. 2, Table 1), we conclude that

**Table 1 Protein–protein interaction and protein–DNA binding activities of GmMYB176 interactome.**

| Candidate | Glyma Id | *In planta* interaction with GmMYB176 | Binding to *GmCHS8-30 bp* promoter fragment |
|---|---|---|---|
| GmMYB176 | Glyma.05G032200.1.p | Yes (homo-dimer) | Yes |
| GmMYB176S29A | Glyma.05G032200.1.p | Yes (homo-dimer) | Yes |
| GmbZIP4 | Glyma.04G222200.1.p | Yes | Yes |
| GmbZIP5 | Glyma.05G122400.1.p | Yes | Yes |
| GmbHLH5 | Glyma.05G134400.1.p | Yes | No |
| GmbHLH7 | Glyma.07G205800.1.p | No | nd |
| GmbHLH15 | Glyma.15G005100.1.p | Yes | No |

*nd* not determined.

GmMYB176 transcriptional complex contains GmbZIP4 and/or GmbZIP5 for *GmCHS8* gene regulation. Investigation of tissue-specific expression of *GmbZIP4*, *GmbZIP5,* and *GmMYB176* revealed that they are co-expressed in roots (Fig. 2d, Supplementary Fig. 1).

**RNAi silencing of *GmbZIP5* reduces isoflavonoid accumulation in hairy roots**. To determine if *GmbZIP4* and/or *GmbZIP5* regulate isoflavonoid biosynthesis, RNAi silencing of both the genes was carried out independently in soybean hairy roots. Transgenic hairy roots were collected from multiple independent transgenic events and combined into several different groups. Each replicate was a group of transgenic soybean hairy roots. Silencing of the target genes *GmbZIP4* and *GmbZIP5* in multiple transgenic hairy roots compared to control tissues were assessed by comparing their transcript levels in GmbZIP4-Si and GmbZIP5-Si, respectively (Fig. 3a). Metabolite extractions were carried out from the same tissues, and total isoflavonoid levels were compared with the control roots. The results revealed that silencing of *GmbZIP5* significantly reduced total isoflavonoid level in GmbZIP5-Si roots compared to the controls (Fig. 3b). However, no change in the total isoflavonoid level was observed in GmbZIP4-Si roots.

**GmMYB176–GmbZIP5 complex is crucial for isoflavonoid biosynthesis in soybean roots**. Previously, we showed that over-expression of GmMYB176 was not able to increase isoflavonoid level in soybean hairy roots[19]. Since GmbZIP4 and GmbZIP5 proteins interacted with the 30 bp *GmCHS8* promoter fragment, we generated two overexpression constructs where *GmMYB176* was translationally fused with *GmbZIP4* (GmMYB176–GmbZIP4) or *GmbZIP5* (GmMYB176–GmbZIP5) using an 18 bp linker DNA (5′-AGCA-CAACATTTCAACCA-3′)[25], and expressed them in soybean hairy roots. Overexpression of *GmMYB176, GmbZIP4,* and *GmbZIP5* in GmMYB176–GmbZIP4 and GmMYB176–GmbZIP5 hairy roots were verified by comparing their expression levels with control roots (Fig. 3c, Supplementary Fig. 2). Analysis of isoflavonoid levels in GmMYB176–GmbZIP4 and GmMYB176–GmbZIP5 hairy roots ($n = 10$) revealed a significant increase in isoflavonoid accumulation in GmMYB176–GmbZIP5 tissues compared to the control hairy roots. No change in isoflavonoid level was observed in GmMYB176–GmbZIP4 hairy roots (Fig. 3d). These results clearly demonstrate that GmbZIP5 is an interacting partner of GmMYB176 for the regulation of isoflavonoid biosynthesis in soybean roots.

To identify metabolites affected by the GmMYB176–GmbZIP5 protein complex, a metabolomic analysis of GmMYB176–GmbZIP5 overexpressing roots were performed, and compared with the control roots using high-resolution mass spectrometry. A total of 8819 and 5508 metabolite features were identified in ESI+ and ESI– modes, respectively (Supplementary Data 3). Of the differentially accumulated metabolite features ($|\log_2|$ fold change >1.0; $p < 0.01$), seven features that corresponded to isoflavonoids were accumulated at 3 to 20 fold higher levels in GmMYB176–GmbZIP5 roots compared to the control (Table 2). Phytoalexin glyceollin and isowighteone levels were 12.7 and 4.7 times higher in GmMYB176–GmbZIP5 roots compared to control suggesting the role of GmMYB176–GmbZIP5 protein complex in disease resistance. We also observed 20.2× higher accumulation of an unknown O-methylhydroxy isoflavone in GmMYB176–GmbZIP5 roots and 5.1 and 4.6× higher accumulation of its glucosyl and malonlyglucosyl conjugates, respectively. This O-methylhydroxy isoflavone has a $m/z$ that corresponds to a chemical formula of $C_{17}H_{14}O_5$. Additionally, MS/MS of this compound revealed a neutral loss of 15.0235, indicative of an O-methyl group. Both of these characteristics coincide with the known compounds afrormosin and alfalone. Afrormosin is a O-methylhydroxy isoflavone found in soybean leaves that are reported to be involved in insect resistance in soybean[26]. The retention time of the O-methylhydroxy isoflavone here was similar to that of afrormosin and the major fragmentation pathway of both compounds was the neutral loss of ·CH₃. However, the major difference was the relative product intensity vis-a-vis the precursor ion (Supplementary Fig. 3). When compared to aflalone, the relative product ion intensity was also dissimilar, as was the retention time. Therefore, although structurally similar, the differentially expressed isoflavonoid found in this study is not afrormosin nor aflalone.

## Discussion

We previously discovered that GmMYB176 regulates isoflavonoid biosynthesis by activating *GmCHS8* gene expression[19]. Furthermore, we demonstrated SGF14 proteins (14-3-3) bind with the phosphorylated GmMYB176 and regulate its shuttling from cytoplasm to the nucleus[24]. The phospho-mutant GmMYB176S29A was unable to interact with SGF14s and localized to the nucleus. Despite that GmMYB176 is necessary for isoflavonoid biosynthesis in soybean roots, it alone is not sufficient for this function[19]. Our main objective in this study was to identify the GmMYB176 interactome and delineate their role in isoflavonoid biosynthesis. Here, we discovered that the unphosphorylated GmMYB176 (GmMYB176S29A) possesses DNA binding activity, interacts with GmbZIP5 in the nucleus (Fig. 2b, c), and that this interaction is critical for isoflavonoid biosynthesis in soybean roots.

Activation of gene transcription by the unphosphorylated transcription factors has been previously reported[27]. In Arabidopsis, three bHLH transcription factors (AKS1, AKS2, and AKS3) in their unphosphorylated state, activate the genes for stomatal opening[28]. Similar to GmMYB176, both phosphorylated and unphosphorylated forms of human Forkhead box O3 (FOXO3) transcription factor bind to the target promoter; however, only the unphosphorylated FOXO3 serves as the activator[29]. Since both GmMYB176 and GmMYB176S29A are able to bind to the *GmCHS8* promoter (Fig. 2b), it is not yet known if the transcriptional complex for its regulation contains phosphorylated or unphosphorylated state of GmMYB176. However, the

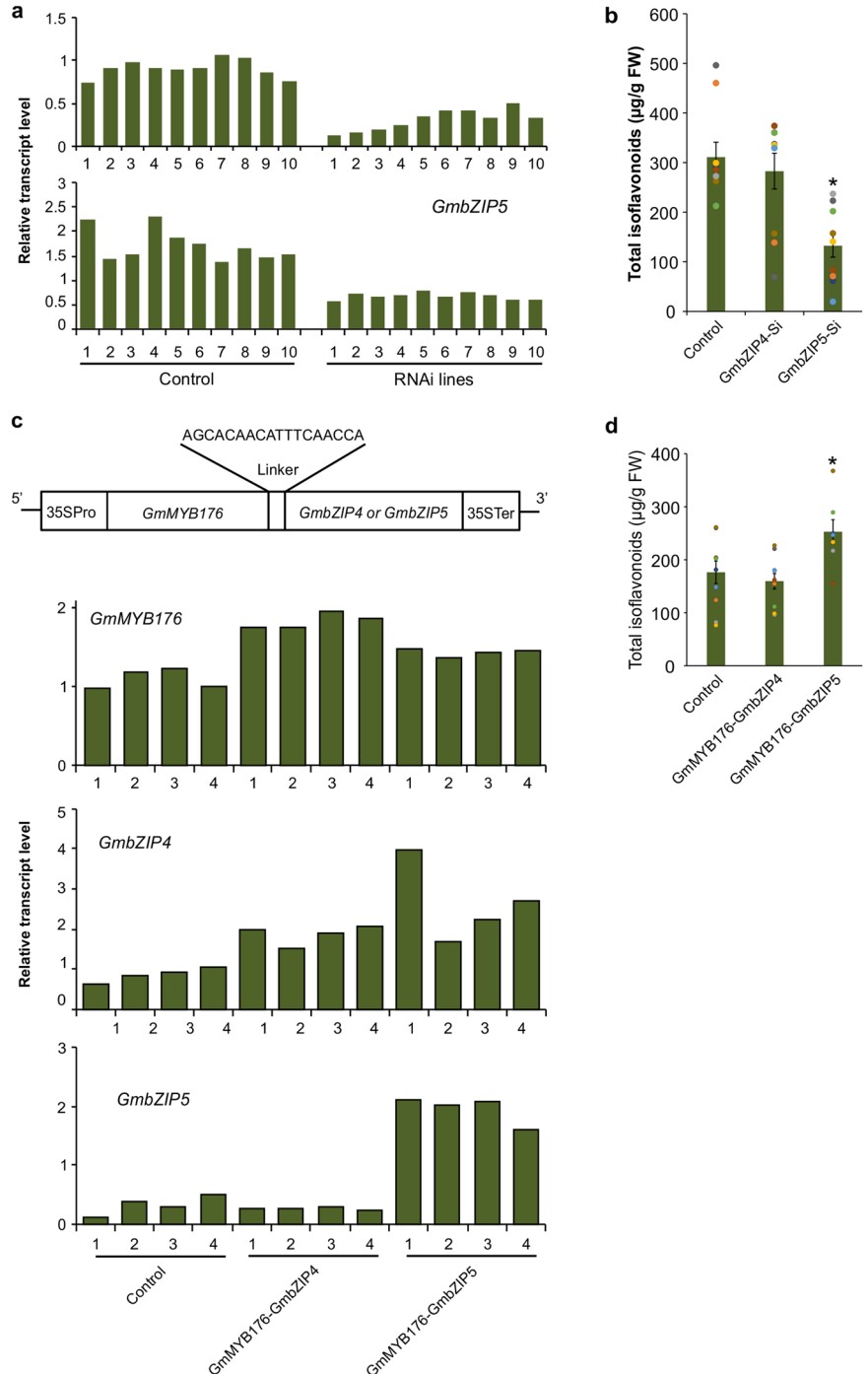

**Fig. 3 Effect of *GmbZIP* silencing and overexpression of *GmMYB176-GmbZIP* fusion complex on isoflavonoid level in soybean hairy roots. a** Accumulation of *GmbZIP4* and *GmbZIP5* transcripts in multiple independent pools of transgenic or control hairy root samples determined by quantitative (q) RT-PCR. Values were normalized against the *CONS4* reference gene. **b** Effect of RNAi silencing of *GmbZIP4* (GmbZIP4-Si) or *GmbZIP5* (GmbZIP5-Si) on isoflavonoid content using hairy roots. Control indicates untransformed hairy roots. Data correspond to mean isoflavonoid levels in ten biological replicates. The asterisk (*) denotes statistically significant expression (one-tail *t* test, *p* < 0.0001). **c** Schematic diagram showing *GmMYB176-GmbZIP4* and *GmMYB176-GmbZIP5* gene fusion for overexpression in soybean hairy roots. Expression levels of the gene fusion were determined by qRT-PCR in multiple independent pools of control or transgenic roots normalized against the reference gene *CONS4*. Control indicates untransformed hairy roots. **d** Effect of overexpression of GmMYB176-GmbZIP4 or GmMYB176-GmbZIP5 on isoflavonoid levels. Data correspond to mean isoflavonoid levels in ten biological replicates. The asterisk (*) denotes statistically significant expression (one-tail *t* test, *p* < 0.01).

strength of YFP signal in the protein–protein interaction between GmbZIP5 and GmMYB176S29A (Fig. 2c) suggests the possibility of unphosphorylated GmMYB176 as the activator. Among the GmMYB176 interactors obtained from the Co-IP experiment, several protein kinase and phosphatase family members were

detected (Supplementary Data 1) and 29 candidates were categorized as "Transcription, DNA-dependent" by GO annotation. We recently reported that alteration in *GmMYB176* expression leads to a substantial alteration in metabolite production stretching beyond the phenylpropanoid pathway in soybean hairy

**Table 2 Differentially accumulated isoflavonoid features in GmMYB176–GmbZIP5 overexpressing soybean hairy roots compared to control roots.**

| Potential metabolite | m/z | RT | Fold change (GmMYB176–GmbZIP5/ control) | Average intensity in GmMYB176–GmbZIP5 | Average intensity in control |
|---|---|---|---|---|---|
| Naringenin glucoside (in source fragment) | 273.0755075 | 2.37 | 3.0 | 46716022.35 | 14154732.97 |
| Glyceollin | 339.1225584 | 3.50 | 12.7 | 181487856.6 | 15409426.1 |
| Malonyl glycitin | 533.1287637 | 2.63 | 3.8 | 25108921.63 | 6483821.772 |
| O-methylhydroxy isoflavone | 299.0911639 | 3.33 | 20.2 | 192451294.2 | 8518735.565 |
| O-methylhydroxy isoflavone glucoside | 461.1439575 | 2.52 | 5.1 | 1417836.2.64 | 2973863.244 |
| O-methylhydroxy isoflavone malonylglucoside | 547.1442021 | 2.96 | 4.6 | 58927604.14 | 13300938.52 |
| [a]Isowighteone | 337.1081469 | 3.10 | 4.7 | 481638.39 | 1029492.57 |

*m/z mass to charge ratio, RT retention time.*
[a]identified in ESI- mode.

roots[20]. Therefore, the possibility of multiple GmMYB176 transcriptional complexes for the regulation of isoflavonoid biosynthetic pathway or pathways beyond isoflavonoids cannot be ruled out.

Overexpression of *GmMYB176* increases the level of only a single (iso) flavonoid precursor, liquiritigenin, suggesting that other isoflavonoid genes were not activated by GmMYB176 alone[20]. RNAi silencing of *GmbZIP5* and co-overexpression of *GmMYB176–GmbZIP5* altered isoflavonoid levels demonstrating a direct influence of GmbZIP5 on isoflavonoid biosynthesis in soybean hairy roots (Fig. 3b, d). Plant bZIPs bind to specific promoter region that contains an ACGT core, such as A-box, C-box, G-box, hybrid C/G-box, or C/A-box motifs[30–32]. Genetic and biochemical analyses in multiple plant species have indicated that bZIP transcription factors act predominantly in stimulus-dependent gene activation[33,34]. There are related flavonoid pathways in other plant species that are synergistically regulated by MYB and bZIP partners. Combinatorial action of a bZIP and a R2R3 MYB factors regulates the light-dependent transcription of the early flavonoid biosynthesis genes such as *CHS*, *CHI*, and *FLS*[33,35]. Both tissue-specific and stress-responsive expression of the French bean *CHS15* expression is regulated by MYB and bZIP-type factors[36,37]. A synergistic regulation of *CHS* gene by a bZIP factor and an LKDKW type R1 MYB, PcMYB, was reported long ago[38], however, GmMYB176 is a SHAQKYF type R1 MYB. Involvement of a transcriptional complex involving and a bZIP and a SHAQKYF type R1 MYB transcription factors has been reported in barley for endosperm-specific gene expression[39]. Nonetheless, the presence of such a complex and its role in plant specialized metabolism was unknown. This is the first evidence of association of a SHAQKYF type R1 MYB and a bZIP (GmMYB176–GmbZIP5) in plant specialized metabolism. The *GmCHS8* promoter contains 12 GmMYB176 binding sites and 5 predicted bZIP binding sites. Despite the presence of multiple GmMYB176 and bZIP binding regions, only the deletion of a 23 bp motif-containing GmMYB176 binding site with a predicted bZIP binding motif within *GmCHS8* alters the promoter activity[19]. The increase in total isoflavonoid level in the hairy roots overexpressing both *GmMYB176* and *GmbZIP5* in the present study confirms that GmbZIP5 is the interacting partner of GmMYB176 (Fig. 3d). Furthermore, co-expression of *GmMYB176* and *GmbZIP5* in soybean roots (Fig. 2d) suggests that GmMYB176–GmbZIP5 complex possibly regulates phytoalexin biosynthesis in soybean roots.

The metabolomics analysis of soybean hairy roots overexpressing *GmMYB176–GmbZIP5* revealed an increase in the accumulation of multiple isoflavonoids such as glyceollins, iso-wighteone, and O-methylhydroxy isoflavones with confirmed or possible roles in plant defense (Table 2). Figure 4 illustrates the proposed biosynthetic pathways of these metabolites where substrate flux is tightly controlled to produce the end products in the pathway. Glyceollins are phytoalexins with key established roles in soybean defense mechanism[40,41] and human health benefits[42]. They are synthesized de novo from the isoflavone daidzein in response to biotic or abiotic stress, and are induced rapidly in the resistant soybean genotypes compared to the susceptible ones[43,44]. Even though, isowighteone, a 3′-prenylgenistein has been found in some plant species with antimicrobial activity[45–47], its presence in soybean was not reported prior to this study. A considerably higher accumulation of O-methylhydroxy isoflavone and its conjugates were also observed in GmMYB176–GmbZIP5 roots. O-methylhydroxy isoflavones such as afrormosin and alfalone have been shown to accumulate in *Medicago truncatula* cell cultures at increased levels upon elicitation[48]. Accumulation of afrormosin has also been linked with insect resistance in soybean[26]. The O-methylhydroxy isoflavone found in this study

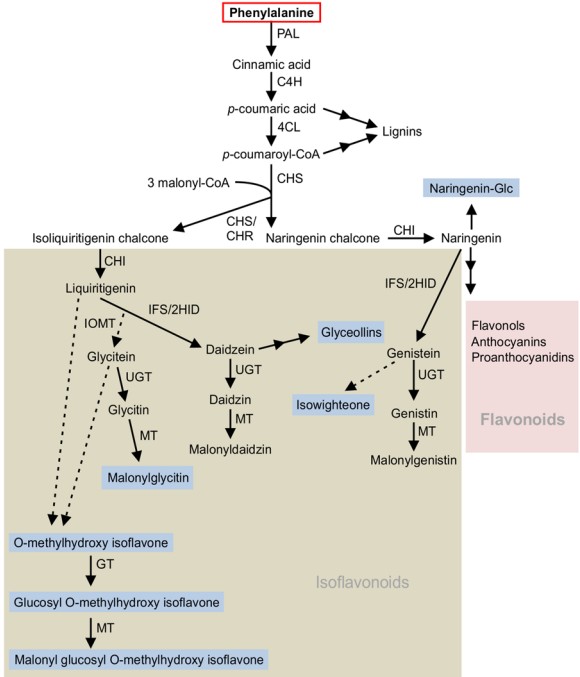

**Fig. 4 Overexpression of *GmMYB176–GmbZIP5* upregulates isoflavonoid biosynthesis.** Multiple arrows indicate multiple steps in the pathway and dotted arrows indicate speculated steps. PAL phenylalanine ammonia-lyase, C4H cinnamate-4-hydroxylate, 4CL 4-coumarate-CoA-ligase, CHS chalcone synthase, CHR chalcone reductase, CHI chalcone isomerase, IFS 2-hydroxyisoflavanone synthase, 2HID 2-hydroxyisoflavanone dehydratase, IOMT isoflavone O-methyltransferase, UGT uridine diphosphate glycosyltransferase, MT malonyltransferase. This pathway is adopted and modified[20]. The blue highlighted metabolites are accumulated at higher level in *GmMYB176–GmbZIP5* overexpression lines compared to control.

is a different metabolite and not afrormosin or aflalone (Supplementary Fig. 2). The dominant fragmentation pathway of both afrormosin and aflalone is the neutral loss of ·CH₃. As the only structural difference between these two compounds is relative positions of the hydroxy and O-methyl groups, the energetics of this fragmentation process are also similar, and thus, the intensity of the *m/z* 284.06 product ion relative to the precursor ion are nearly identical (Supplementary Fig. 2). In contrast to both afrormosin and aflalone, the methylhydroxy isoflavone of this study has a much higher product ion intensity relative to the precursor ion. This was confirmed by carefully performing the MS/MS experiment at multiple collision energies. Relative intensities of product ions are linked to the energetics and entropics of the specific unimolecular dissociation[49]. The increased intensity of the *m/z* 284.04 product ion of the methylhydroxy isoflavone found in this study suggests that although it is highly structurally similar to afrormosin and aflalone, there is likely a key, positional difference. Its similarity to afrormosin or aflalone suggests its role in plant defense and the fact that it and its conjugates are at substantially higher levels in GmMYB176–GmbZIP5 roots compared to controls necessitates the discovery of the role of this specialized metabolite in the plant.

Taken together, we show here that a combined action of GmMYB176–GmbZIP5 is critical to activate isoflavonoid biosynthesis in soybean roots. This study also reports the identification of the three unique isoflavonoids—O-methylhydroxy isoflavone, glucosyl O-methylhydroxy isoflavone, and malonyl glucosyl O-methylhydroxy isoflavone in soybean roots. Since root

isoflavonoids affect nodulation and resistance against diseases due to soil-borne pathogens such as *P. sojae*, identification of the structure and function of the O-methylhydroxy isoflavone identified in this study may provide information on novel biological or biochemical phenomena. Our findings could be utilized to develop biotechnological or traditional breeding strategies to manipulate isoflavonoid levels either for the enhancement of nutritional value or for protection against plant diseases in soybean or other agronomical important crops.

## Methods

**Plant materials and growth conditions**. *Nicotiana benthamiana* plants were grown in Pro-Mix BX Mycorrhizae™ soil (Riviere-du-Loup, Canada) under 16-h light at 25 °C and 8-h dark at 20 °C cycles with 60–70% relative humidity and the light intensity of 80 μmol photons/m²/s[50].

To obtain soybean cotyledons for hairy root transformation, soybean (*Glycine max* L. Merr.) cv. Harosoy63 seeds were surface sterilized with 70% ethanol (v/v) containing 3% H₂O₂ (v/v) for 2 min and then rinsed with sterile water prior to planting. Seeds were planted in vermiculite and grown for 6 days in a growth chamber under the condition described earlier.

**Plasmid construction**. All plasmid constructions were performed using Gateway technology (Invitrogen, USA). For the PPI study, genes of interest (GOI) were cloned into Gateway destination vectors, pEarleyGate201-YN and pEarleyGate202-YC[51]. For Co-IP, the GmMYB176 was cloned into pEarleyGate101 and pEarleyGate104[52] to obtain pEG101-GmMYB176-YFP and pEG104-YFP-GmMYB176, respectively. For RNAi, *GmbZIP* genes were cloned into pK7GWIWG2D (II). For overexpression, *GmMYB176–GmbZIP* gene fusion was created using an 18 bp linker (AGCACAA CATTTCAACCA) by fusion PCR using the primers listed in Supplementary Data 4. The PCR products were cloned into the Gateway vector pK7WG2D. All the plasmid constructs were transformed individually into *Agrobacterium tumefaciens* GV3101 by electroporation.

For Y1H assay, 30 bp GmCHS8 promoter fragments in three tandem repeats (107 bp) were synthesized (Supplementary Table 4) and cloned into a pAbAi vector to obtain p30bpTR-AbAi. The prey GOI were cloned into pGADT7 using Gateway technology (Invitrogen, USA).

**Generation of soybean hairy roots**. Six-day-old soybean cotyledons were harvested for hairy root generation. *A. rhizogenes* K599 containing GOI in destination vector was inoculated into soybean cotyledons[53]. Transgenic hairy roots were selected 20–30 days post inoculation, using a Leica MZ FL III fluorescence stereo microscope with a YFP filter (excitation 510/520 nm; barrier filter 560/540 nm) and flash frozen and stored at −80 °C until use.

**Subcellular localization and BiFC assay**. The leaves of 4–6 week old *N. benthamiana* plants were infiltrated with *A. tumefaciens* culture containing GOI in the appropriate destination vectors[54]. To verify PPI, constructs in pEarleyGate201-geneA and pEarleyGate202-geneB were co-transformed in a 1:1 mixture. Confocal microscopy was carried out 48 h post infiltration using a Leica TCS SP2 inverted confocal microscope. The excitation and emission of YFP were 514 nm and 530–560 nm, respectively.

**Protein extraction, Co-IP assay, *in-gel* digestion, and LC–MS/MS**. Proteins were extracted from soybean hairy roots overexpressing GmMYB176-YFP or YFP-GmMYB176, and co-IP was performed as previously described[21]. The GmMYB176-YFP and YFP-GmMYB176 fusion proteins were identified by Western blot analysis after sequential incubation of the blot with the Living Colors® A. v. (anti-GFP) Monoclonal Antibody (Clontech, USA) at a dilution of 1:3000 followed by the HRP-conjugated goat anti-mouse secondary antibody (Pierce, USA) at a dilution of 1:5000. HRP detection was performed using Super Signal West Femto (Thermo Scientific, Canada).

The protein eluates (1 μg) were separated by SDS-PAGE followed by silver staining using ProteoSilver kit (Sigma, USA). Protein bands were excised from gels and destained with 30 mM K₃[Fe(CN)₆] and 100 mM Na₂S₂O₃ solution followed by in-gel trypsin digestion using a MassPREP automated digester station (PerkinElmer, USA). Peptides were extracted using a buffer containing 20 mM Tris-HCl (pH 7.5), 150 mM NaCl, and analyzed by LC–MS/MS (Waters NanoAcquity UPLC coupled with Thermo Orbitrap Elite ETD) in Biological Mass Spectrometry Laboratory (London Regional Proteomics Centre, Canada). MS/MS data were analyzed by MassLynx 4.1 software with Mascot (http://www.matrixscience.com) and compared against the soybean protein database in Phytozome[22,55] with parameters of monoisotopic peptide mass adopted, mass window ranged from 1 kDa to 100 kDa, mass tolerance set as 50 ppm and allowance of one missed cleavage. Proteomics data were deposited to the ProteomeXchange Consortium via the PRIDE[56] partner repository with the dataset identifier PXD023931.

**Y1H assay**. Y1H assays were carried out by following the Matchmaker® Gold Yeast One-Hybrid Library Screening System User Manual (Clontech, USA). The recombinant plasmid p30bpTR-AbAi was transformed into yeast (Y1H Gold strain) using the Yeastmaker™ Yeast Transformation System 2 (Clontech, USA) and grown on SD/-Ura media at 30 °C for 3 days. The transformed colonies were screened by colony PCR using Matchmaker Insert Check PCR Mix 1 (Clontech, USA). Y1H assays were performed by following the Matchmaker Y1H user manual (Clontech, USA). The minimum inhibitory concentration of aeuro-obasidin A (AbAi) for yeasts carrying 30bpTR promoter bait was 150 ng/mL. The prey constructs (pGADT7-GOI) were transformed into yeast carrying a promoter bait fragment, plated on SD/-Leu/AbAi150, and incubated at 30 °C for 3–5 days.

**Quantitative RT-PCR analysis**. RNA was isolated from soybean hairy roots using RNeasy Plant Mini Kit (Qiagen, USA). Total RNA (1 μg) was used for cDNA synthesis using the ThermoScript™ RT-PCR Systems (Invitrogen, USA). Gene-specific primer sequences for qPCR are listed in Supplementary Data 4. All reactions were performed in three technical replicates, and the expression was normalized to the reference gene CONS4[57]. The data were analyzed using Bio-Rad CFX Maestro (Bio-Rad, USA) (Supplementary Data 5).

**Metabolite extraction, HPLC, and LC–MS/MS analysis**. Total isoflavonoid extraction and HPLC analysis from soybean hairy roots were performed as previously described[9]. For metabolomics analysis, frozen hairy roots were ground with liquid nitrogen and extracted in methanol:water (80:20, v/v). The samples were sonicated on an ice water bath for 15 min followed by centrifugation at $11,000 \times g$ for 10 min at ambient temperature. The supernatant (350 μL) was dried under nitrogen gas. The dried pellet was dissolved in 200 μL of 50% methanol containing 10 μg caffeine as an internal standard and filtered through a 0.45 μm syringe filter (Millipore, United States).

Samples (5 μL) were injected to an Agilent 1290 HPLC coupled to a Q-Exactive Quadrupole Orbitrap mass spectrometer (ThermoFisher Scientific, United States) for high-resolution LC–MS analysis as described previously with some modification[20]. Heated electrospray ionization (HESI) conditions used are as follows; spray voltage, 3.9 kV (HESI+), −3.5 kV (HESI−); capillary temperature, 400 °C; probe heater temperature, 450 °C; sheath gas, 17 arbitrary units; auxiliary gas, eight arbitrary units; and S-Lens RF level, 45. Single stage, full MS at 140,000 resolutions, full mass scans between the range of $m/z$ 100 to 1000 in both positive and negation ionization were used for differential analysis. Automatic gain control (AGC) target and maximum injection time (IT) were $5 \times 10^5$ and 512 ms, respectively. For metabolite feature identification, data-dependent acquisition (DDA) mode experiments were used for a representative sample from each treatment. The DDA methods used identical HESI conditions and comprised of a full MS scan at 17,500 resolution between $m/z$ range of 100 to 1000, AGC target of $1 \times 10^6$ and maximum IT of 64 ms. The top 15 most intense ions above a threshold of $1 \times 10^4$ were sequentially selected for MS/MS using a 1.2 $m/z$ isolation window, normalized collision energy (NCE) of 35, and excluded from MS/MS for 5 s. Compounds were identified as previously described[20]. For the metabolomics analysis and alignment of the detected peaks, the XCMS package in R was used as described by Gracia et al.[58] with the addition of diffreport-method to create a summary report. Compounds that showed differential accumulation were chosen for further identification through Xcalibur software.

**Statistics and reproducibility**. Statistical analyses were performed using Microsoft Office Excel. Values were expressed as means ± standard error (SE). Statistically significant between two samples were determined by comparing means using Student's $t$ test (one-tail, unpaired) with $P < 0.01$. All experiments were performed at least four times with similar results.

**Reporting summary**. Further information on research design is available in the Nature Research Reporting Summary linked to this article.

## Data availability

All data generated or analyzed during this study are included in this published article either in the Source Data file, via respective repository entry, or Supplementary Information files and are available from the corresponding author on reasonable request and are available from the corresponding author on reasonable request. The metabolomics LC–MS data can be accessed from Metabolomics workbench study ST001634 (https://www.metabolomicsworkbench.org/data/DRCCMetadata.php?Mode=Study&StudyID=ST001634&StudyType=MS&ResultType=5). The Co-IP mass spectrometry (MS) proteomics data have been deposited to the ProteomeXchange Consortium via the PRIDE partner repository (Identifier PXD023931).

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

## Acknowledgements

We thank Ling Chen, Nishat Shayala Islam, Alex Molnar, and Michelle Bargel (London Research and Development Centre, AAFC) for technical assistance. Kristina Jurcic (MALDI Mass Spectrometry Facility, UWO, Canada) and Paula Pittock (Biological Mass Spectrometry Laboratory, UWO, Canada) for help with proteomic work. This work was supported by the Natural Sciences and Engineering Research Council of Canada's Discovery Grant (385922-2011RGPIN) and Agriculture and Agri-Food Canada's Abase Grant (J-000151) to S.D.

## Author contributions

A.K.A.V. performed all the experiments, collected and analyzed data, and prepared draft manuscript. T.M. and J.B.R. performed metabolomics experiment, analyzed data, and contributed to manuscript preparation. S.D. conceived and designed experiments, supervised all aspects of the project, and prepared the final draft manuscript.

## Competing interests

The authors declare no competing interests.
