## [Peer Review File · Communications Biology]

Reviewers' comments:

Reviewer #1 (Remarks to the Author):

This manuscript claims:

- 1) GmbZIP5 interacts with GmMYB176.
- 2) Down-regulation of GmbZIP5 reduced the isoflavonoid level in soybean hairy roots.
- 3) Co-overexpression of GmMYB176 and GmbZIP5 enhanced some isoflavonoid level in soybean hairy roots.

The conclusion was based on soybean hairy root transformation. How about this approach compared to soybean transformation? And how about the effect of GmMYB176 and GmbZIP5 in seed isoflavonoid accumulation?

This manuscript has some parts that needs to be polished. For example,
"337 Y1H assays

338 Y1H assays were carried out by following the Matchmaker® Gold Yeast One-Hybrid Library
339 Screening System User Manual (Clontech, USA). "---Add some details for this experiment.

"429 This pathway is adaopted and modified(19). The blue highlighted metabolites are accumulate at
430 higher level in GmMYB176-GmbZIP5 overexpression lines compared to control."--- adopted, are
accumulated.

Reviewer #2 (Remarks to the Author):

The manuscript by Vadivel et al., is to identify the GmMYB176 interactome for the regulation of isoflavonoid biosynthesis in soybean. The study reported is well established and critical experimental design has been put forward to create the link between GmMYB176 and GmbZIP5 to see the level of isoflavonoids in soybean hairy roots.

Overall the study is very compact and significant data has been presented to support the hypothesis. I have some minor concerns regarding the experimental set-up and in results section.

1. Line 42 – “Soybean contains fourteen GmCHS genes”, can author add a statement whether these genes are also showing some changes during abiotic stress. Any one of the gene is studied widely or not. As far I have seen the literature it can?
2. Line 62 – “Furthermore, detailed functional analysis of GmMYB176 revealed that it requires additional factor(s)”, elaborate which additional factors authors are hinting towards.
3. Line 283-289 – “Plant materials and growth conditions” - cite a reference for this experimental setup.
4. Line 305 – “Six-day-old soybean cotyledons” – what was the height of the seedling after 6 days. Did you measure the height?
5. For metabolomics analysis which software was used to identify the peaks and what was the injection volume in the LC-MS.
6. Line 94-100 – All the proteins which authors are describing are indeed proteins, I hope they are not post-transcriptional regulators.
7. Line 260 - O-methylhydroxy isoflavone is a new metabolite. Could you give some more details about this metabolite? In which others species this was discovered earlier.
8. Line 278- spelling error. isoflavpnoid
9. Conclusion is very abrupt. Could you elaborate on it?
10. Fig 2b – Sharpness in the figure is missing. Especially the last slides of GmMYB176S29A.

Response to Reviewers' comments:

Reviewer #1

This manuscript claims:

- 1) GmbZIP5 interacts with GmMYB176.
- 2) Down-regulation of GmbZIP5 reduced the isoflavonoid level in soybean hairy roots.
- 3) Co-overexpression of GmMYB176 and GmbZIP5 enhanced some isoflavonoid level in soybean hairy roots.

The conclusion was based on soybean hairy root transformation. How about this approach compared to soybean transformation? And how about the effect of GmMYB176 and GmbZIP5 in seed isoflavonoid accumulation?

Response: Hairy root transformation is a well-established transgenic model system for the functional characterization of genes involved in the traits that are present in roots in many plant species including soybean¹⁻³. Soybean transformation is very time consuming and takes over a year to get T1 plants. Since both soybean roots⁴ and hairy roots⁵ contain large amount of isoflavonoids, we decided to use hairy root system. The pathway gene expressions and isoflavonoid content in hairy roots and soybean roots are comparable and transferable. Using stable soybean transformation system will yield the same outcome as the hairy root system for isoflavonoids as the pathway is active in both the tissues. Overexpression of GmMYB176 and GmbZIP5 in seed will definitely increase the levels of isoflavonoid accumulation in seed tissue. However, under normal developmental condition, both GmMYB176 and GmbZIP5 in developing seeds are very low (new Supplementary Fig.1), therefore, other transcription factors such as GmMYB29¹ may be involved in seed-specific isoflavonoid synthesis. However, total isoflavonoid content in soybean seeds are due to the synthesis within the seed tissue as well as transport from other plant parts⁴. Therefore, root isoflavonoid level due to the action of GmMYB176 and GmbZIP5 also contribute to the total isoflavonoid accumulation in soybean seeds.

This manuscript has some parts that needs to be polished. For example,

"337 Y1H assays

338 Y1H assays were carried out by following the Matchmaker® Gold Yeast One-Hybrid Library
339 Screening System User Manual (Clontech, USA). "---Add some details for this experiment.

Response: As per the Reviewer's suggestion, we have added the experimental details in this section. Please refer to Lines 343 to 352.

"429 This pathway is adopted and modified(19). The blue highlighted metabolites are accumulate at
430 higher level in GmMYB176-GmbZIP5 overexpression lines compared to control."--- adopted,
are accumulated.

Response: The identified errors have been corrected. Thank you.

Reviewer #2:

The manuscript by Vadivel et al., is to identify the GmMYB176 interactome for the regulation of isoflavonoid biosynthesis in soybean. The study reported is well established and critical experimental design has been put forward to create the link between GmMYB176 and GmbZIP5 to see the level of isoflavonoids in soybean hairy roots.

Overall the study is very compact and significant data has been presented to support the hypothesis. I have some minor concerns regarding the experimental set-up and in results section.

1. Line 42 – “Soybean contains fourteen GmCHS genes”, can author add a statement whether these genes are also showing some changes during abiotic stress. Any one of the gene is studied widely or not. As far I have seen the literature it can?

Response: As suggested by the Reviewer, we have added a citation showing effect of abiotic stress on CHS gene expression. The environmental stimuli we have mentioned here includes both biotic and abiotic stresses. Among the *GmCHS* members *CHS7* and *CHS8* are widely studied for their roles in isoflavonoids biosynthesis and seed color. This has been mentioned in the revised manuscript. Please refer to lines 43 to 47.

2. Line 62 – “Furthermore, detailed functional analysis of GmMYB176 revealed that it requires additional factor(s)”, elaborate which additional factors authors are hinting towards.

Response: As per the Reviewer’s suggestion, we have elaborated on the additional factors. Please refer to lines 65 to 67.

3. Line 283-289 – “Plant materials and growth conditions” - cite a reference for this experimental setup.

Response: As suggested by the Reviewer, we have added a citation for the experimental set up. Please refer to line 290.

4. Line 305 – “Six-day-old soybean cotyledons” – what was the height of the seedling after 6 days. Did you measure the height?

Response: Since the same soybean cultivar Harosoy 63 was grown at the same time under the same condition, we did not measure the height of the seedlings. But they were of approximately 5-6 cm. If required, we can grow them again under the same growth condition and measure the height.

5. For metabolomics analysis which software was used to identify the peaks and what was the injection volume in the LC-MS.

Response: The XCMS package in R was used to align the detected peaks as described by Gracia et al⁶ with the addition of diffreport-method to create summary report. Compounds that showed up/down regulation were chosen for further identification through Xcalibur software. The injection volume was 5 µL. This information has been added in the manuscript text. Please refer to lines 383 to 387 and line 369.

6. Line 94-100 – All the proteins which authors are describing are indeed proteins, I hope they are not post-transcriptional regulators.

Response: Yes, they are proteins.

7. Line 260 - O-methylhydroxy isoflavone is a new metabolite. Could you give some more details about this metabolite? In which others species this was discovered earlier.

Response: Since the identity of the O-methylhydroxy isoflavone in our study is unknown, its presence in other plant species is not known. Afformosin and alfalone are structurally similar compounds and are commercially available, therefore, using LC-MS, we confirmed that O-methylhydroxy isoflavone identified in our study is neither afformosin nor alfalone. It has been a challenge to determine the position of all the OHs without performing NMR. This has been explained in lines 261 to 275.

8. Line 278- spelling error. Isoflavpnoid.

Response: corrected.

9. Conclusion is very abrupt. Could you elaborate on it?

Response: As suggested, we have elaborated the conclusion. Lines 277 to 282.

10. Fig 2b – Sharpness in the figure is missing. Especially the last slides of GmMYB176S29A.

Response: corrected.

References

- 1 Chu, S. *et al.* An R2R3-type MYB transcription factor, GmMYB29, regulates isoflavone biosynthesis in soybean. *PLoS Genet.* **13**, e1006770, doi:10.1371/journal.pgen.1006770 (2017).
- 2 Jun, J. H., Liu, C., Xiao, X. & Dixon, R. A. The transcriptional repressor MYB2 regulates both spatial and temporal patterns of proanthocyanidin and anthocyanin pigmentation in *Medicago truncatula*. *Plant Cell* **27**, 2860-2879, doi:10.1105/tpc.15.00476 (2015).
- 3 Li, P. *et al.* Regulation of anthocyanin and proanthocyanidin biosynthesis by *Medicago truncatula* bHLH transcription factor MtTT8. *New Phytol.* **210**, 905-921, doi:10.1111/nph.13816 (2016).
- 4 Dhaubhadel, S., McGarvey, B. D., Williams, R. & Gijzen, M. Isoflavonoid biosynthesis and accumulation in developing soybean seeds. *Plant Mol. Biol.* **53**, 733-743 (2003).
- 5 Yi, J. *et al.* A single-repeat MYB transcription factor, GmMYB176, regulates CHS8 gene expression and affects isoflavonoid biosynthesis in soybean. *Plant J.* **62**, 1019-1034, doi:10.1111/j.1365-313X.2010.04214.x (2010).
- 6 Garcia, E. J. *et al.* Metabolomics reveals chemical changes in *Acer saccharum* sap over a maple syrup production season. *PLoS One* **15**, e0235787, doi:10.1371/journal.pone.0235787 (2020).

REVIEWERS' COMMENTS:

Reviewer #1 (Remarks to the Author):

Manuscript is improved.

Reviewer #2 (Remarks to the Author):

All my points were carefully addressed and the manuscript was significantly improved.